# GThinker: Towards General Multimodal Reasoning via Cue-Guided Rethinking

## Abstract

Despite recent advances in multimodal reasoning, Multimodal Large Language Models (MLLMs) still underperform on complex vision-centric reasoning tasks compared to their strong capabilities in language-based reasoning. This performance gap stems from a critical asymmetry in their reasoning processes: while MLLMs excel at iterative reflection and correction in textual contexts, they tend to uncritically accept their initial visual interpretations and rarely revise them, even when these cues lead to logical inconsistencies. To overcome this shortcoming, we introduce **GThinker**, a general-purpose reasoning MLLM that unifies robust textual reasoning with a novel, adaptive visual rethinking capability. GThinker first introduces Cue-Rethinking, a flexible reasoning pattern that not only grounds reasoning in visual cues but also strategically triggers rethinking of these cues to resolve visual inconsistency for solid reasoning. To cultivate this adaptive capability across domains, we further design a two-stage training pipeline, including the pattern-guided cold start with the judge-guided selective training and incentive reinforcement learning. Furthermore, we construct GThinker-11k to support the training, a dataset containing 7K cue-annotated chain-of-thought data and 4K diverse reinforcement samples, using the designed iterative multimodal annotation pipeline. Extensive experiments demonstrate that GThinker achieves 81.5% on the challenging comprehensive multimodal reasoning benchmark $M^3$CoT, surpassing the latest O4-mini model. It also shows an average improvement of 2.1% on general scenario multimodal reasoning benchmarks, while maintaining on-par performance in mathematical reasoning compared to counterpart advanced reasoning models.

## 1 Introduction

Open-source Multimodal Large Language Models (MLLMs) (Li et al., 2024a; Wu et al., 2024; Zhu et al., 2025; Wu et al., 2025; Team et al., 2025) have made significant strides across a wide range of tasks. Leading models like Qwen2.5-VL (Bai et al., 2025) now rival closed-source counterparts such as GPT-4o (Hurst et al., 2024) in performance. These advances have benefited in part from the adoption of chain-of-thought (CoT) techniques (Lu et al., 2022a; Yao et al., 2023; Wei et al., 2022), especially in mathematics and science. With the emergence of OpenAI's O1 model (Jaech et al., 2024), several studies (Yao et al., 2024; Xu et al., 2024; Thawakar et al., 2025) have sought to mimic such human-like CoT reasoning capabilities in the multimodal reasoning domain to enhance models' performance on complex tasks. DeepSeek-R1 (Guo et al., 2025) further introduces a new perspective, showing that Reinforcement Learning with Verified Rewards (RLVR) can awaken such CoT reasoning, with promising results (Meng et al., 2025; Yang et al., 2025; Chen et al., 2025a) in multimodal reasoning tasks involving science and mathematics.

Beyond mathematics and science, multimodal reasoning in general scenarios, which often involves visual cues and related commonsense, still remains under-explored. Unlike math and science tasks, where strict logical structures and unique answers allow models to benefit from consistent reflection on textual derivations, general multimodal reasoning is inherently diverse and more reliant on visual interpretation and deduction. This diversity makes it first challenging to summarize a fixed CoT pattern or design an effective Process Reward Model (PRM), limiting the effectiveness of structured reasoning (Yao et al., 2024; Xu et al., 2024) and Multimodal PRMs (Wang et al., 2025; Liu et al., 2024a). Moreover, we observe a critical asymmetry in the reasoning process of current reasoning

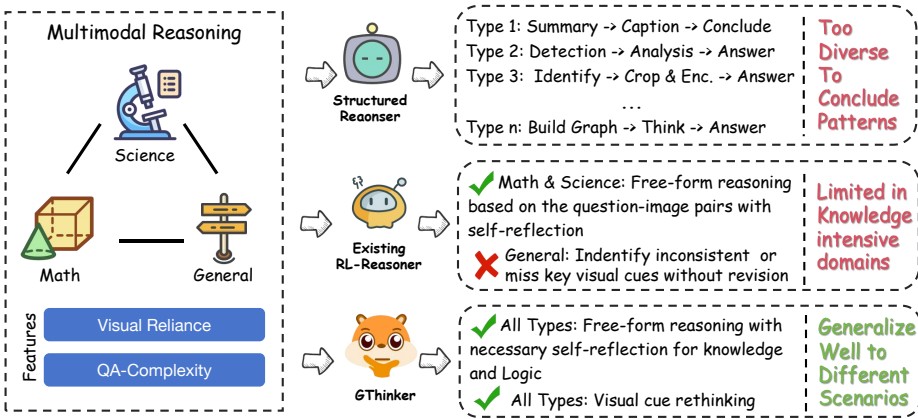

Figure 1: Multimodal reasoning methods comparison across scenarios. Multimodal Reasoning in different domains is featured with visual reliance and high question complexity, making it a challenging task. Different from previous methods, GThinker utilizes **near free-form** thinking for different types of questions instead of a fixed structure form and enables general scenario reasoning accuracy with **designed visual cue rethinking**.

MLLMs trained with verifiable rewards: while they can iteratively reflect on and correct textual context, they tend to uncritically accept initial visual interpretations and rarely revise them even when inconsistencies arise. As summarized in Figure 1, existing RLVR reasoning models often fail to revisit or update visual cues once an initial interpretation is formed, and when confronted with plausible yet inconsistent intermediate states, they continue along the current path to an answer rather than re-examining those cues. This form of visual rethinking is fundamentally different from text-centric reflection (Shah et al., 2025), which cannot be reliably induced by RLVR alone, nor captured by a fixed CoT or PRM template. These limitations point to the need for a mechanism that unifies robust textual logic with adaptive re-interpretation of visual cues.

To address these challenges, we propose GThinker, a novel reasoning MLLM, which unifies robust textual logic with a novel, adaptive visual rethinking capability to excel in multimodal reasoning across general scenarios, mathematics, and science. First, we introduce a new visual cues-driven multimodal reasoning pattern called Cue-Rethinking to make reasoning as natural as text-centric reflection. Unlike prior approaches (Xu et al., 2024; Thawakar et al., 2025) that define structured CoT formats, Cue-Rethinking only requires the reasoning process to be strictly grounded in visual cues without enforcing a fixed format. After completing an initial reasoning chain, the model rethinks the interpretations and inferences based on visual content to correct inconsistencies and arrive at the correct answer. Building on this pattern, we propose a two-stage training pipeline to enable robust multimodal reasoning across domains. We begin with a pattern-guided cold start that not only teaches the model this reasoning pattern across tasks but also cultivates adaptive decisions about when to rethink via judge-guided selective training. Then, we further employ an incentive RL stage to let the model explore optimal strategies for solving diverse problems across domains. To support training, we further develop a multimodal iterative annotation pipeline based on the latest advancing multimodal models like O3 (OpenAI, 2025) and construct GThinker-11k, comprising 7K cold-start data with high-quality annotated reasoning paths and 4K reinforcement learning samples, filling a key gap in multimodal reasoning data for general scenarios.

We implement GThinker based on the advanced open-source MLLM Qwen-VL 2.5–7B and conduct extensive experiments to rigorously evaluate its effectiveness. We first benchmark GThinker against both open- and closed-source models on M$^3$CoT (Chen et al., 2024a), a challenging and comprehensive multimodal reasoning dataset spanning science, general commonsense, and mathematics. For broader validation, we include general-domain benchmarks such as MMStar (Chen et al.) and RealWorldQA (xAI, 2024), as well as science and math-focused benchmarks including MMMU-Pro (Yue et al., 2024), MathVision (Wang et al., 2024a), and MathVista (Lu et al.). GThinker demonstrates strong performance across all domains, achieving 81.5% on M$^3$CoT—surpassing the advanced O4-mini model. On MMStar and RealWorldQA, GThinker achieves the improvement of

2.5% and 1.6%, respectively. Additionally, it performs competitively on science and math benchmarks with 40.7% on MMMU Pro and 72.7% on MathVista, matching or outperforming recent RL-enhanced approaches, further validating its effectiveness.

## 2 RELATED WORK

### 2.1 STRUCTURED MULTIMODAL CHAIN-OF-THOUGHT REASONING

Structured Multimodal Chain-of-Thought (MCoT) reasoning builds on the Chain-of-Thought (CoT) paradigm (Wei et al., 2022), extending it to multimodal tasks using step-by-step reasoning (Lu et al., 2022a; Zhang et al., 2023). Many approaches enhance this framework with structured designs (Zheng et al., 2023; Liu et al., 2024b; Mitra et al., 2024) and further improvements such as fine-grained visual grounding, context integration, or tool use (Jia et al., 2024; Gao et al., 2024; Luan et al., 2024; Wu & Xie, 2024; Shao et al., 2024a; Li et al., 2024b; Bigverdi et al., 2024). However, these methods are often task-specific—e.g., CCoT (Mitra et al., 2024) for compositional reasoning, LLaVA-Aurora (Bigverdi et al., 2024) for spatial reasoning—and lack robustness across diverse scenarios. Recently, slow-thinking paradigms (Jaech et al., 2024; Team, 2025; Qin et al., 2024) have been proposed to improve reasoning depth. Enhanced MCoT variants like LLaVA-CoT (Xu et al., 2024), Virgo (Du et al., 2025), and Mulberry (Yao et al., 2024) leverage long-chain generation, tree search, and self-reflection. Yet, they remain confined to structured, logic-heavy tasks and are difficult to generalize to broader settings. In contrast, GThinker adopts a free-form, cue-based thinking paradigm with further visual cue-based rethinking, moving beyond rigid structures to support open-domain multimodal reasoning. This design enables generalization across tasks without sacrificing interpretability or performance.

### 2.2 MULTIMODAL REASONING WITH REINFORCEMENT LEARNING

3. Reinforcement learning (RL) has become a powerful tool to align MLLMs and mitigate hallucinations (Sun et al., 2023b; Yu et al., 2024; Zhang et al., 2024b; Sun et al., 2023a; Li et al., 2023; Zhang et al., 2025), and is now being explored to improve multimodal reasoning. Early works like LLaVA-Reasoner (Zhang et al., 2024a), MPO (Wang et al., 2024c), and Insight-V (Rafailov et al., 2023) emphasize supervised or preference signals to teach correctness, limiting robustness and scalability for more complex scenarios. A shift emerged with DeepSeek-R1 (Guo et al., 2025), which showed that outcome-based rewards, without fine-grained annotations, can drive reasoning through self-verification and reflection. Follow-up works (Chen et al., 2025b; Yang et al., 2025; Meng et al., 2025; Chen et al., 2025a; Team et al., 2025; Peng et al., 2025) expand this idea to the multimodal domain, leveraging different verifiable reward functions or data sampling approaches to improve math and science reasoning. However, these methods chiefly strengthen reflection over textual steps and seldom encourage re-examination of visual evidence, which limits performance on general multimodal tasks with ambiguous cues. In contrast, we make adaptive visual rethinking a main objective together with textual thinking by a cue-rethinking pattern that teaches not only how to rethink but also when to invoke it, enabled by a two-stage training scheme.

## 3 METHODOLOGY

In this section, we provide a comprehensive description of the novel multimodal reasoning model GThinker as depicted in Figure 4. In §3.1, we first present the Cue-Rethinking Pattern, a core component built on free-form thinking to provide visual cue-driven guidance for multimodal reasoning across scenarios. Then, in §3.2, we describe Pattern-Guided Cold Start, in which we train the model with pattern-guided supervised fine-tuning to learn how to think and when to rethink for different scenarios. Finally, we introduce Incentive Reinforcement Learning to generalize the multimodal reasoning capabilities of the model across diverse scenarios in §3.3.

### 3.1 CUE-RETHINKING PATTERN

Existing CoT reasoning methods(Xu et al., 2024; Thawakar et al., 2025) often rely on fixed, structured thinking chains tailored to specific tasks. While effective in targeted domains, their perfor-

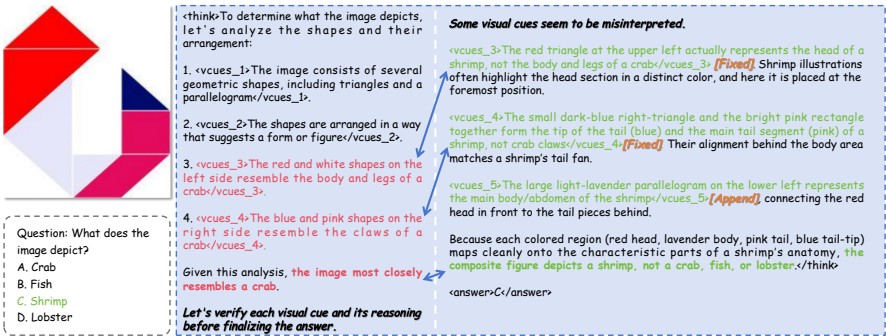

Figure 2: Constructed Data Example with Cue-Rethinking. The visual cues in red are flawed ones, while the green indicates the visual cues are revised or appended.

mance tends to drop sharply when applied to more general or unfamiliar scenarios. RLVR models offer more flexibility, but they also fall short in general settings that require grounded, visually informed interpretations and deductions. To tackle this challenge, we introduce the Cue-Rethinking Pattern, a thinking mechanism that enables flexible CoT reasoning capable of natural rethinking on visual cues with near free-form textual thinking.

As shown in Figure 3, the process has three stages: initial reasoning, cue-rethinking trigger, and cue-based rethinking. In the initial stage, the model may use any previously learned textual reasoning strategy—e.g., step-by-step deduction, reflection, or knowledge-driven logic—without structural constraints. The only requirement is to ground its reasoning in visual evidence and explicitly mark the referenced cues using the format <vcues_*> </vcues_*> (* denotes the

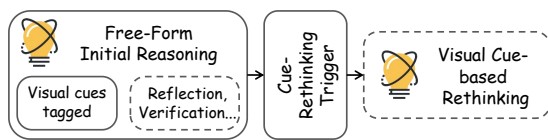

Figure 3: Toy example of the Cue-Rethinking Pattern. The dashed line indicates generation on demand.

cue index). This lightweight constraint preserves flexibility while highlighting the visual cues explicit from the outset, and provides clear anchors for subsequent rethinking.

After the initial reasoning, the model self-triggers a prompt for cue-based rethinking ( *e.g., "Let's check each visual cue and the corresponding reasoning before the final answer"*). We intentionally avoid immediate rethinking after cue identification to maintain natural reasoning flow and preserve global context. The model then revisits all marked visual cues, checks for inconsistencies, and, if needed, revises the cues and updates the associated reasoning before producing the final answer, as indicated on the right of Figure 2. This design remains compatible with well-established text-centric reasoning (*e.g.*, knowledge-driven logic, reflective deduction) while explicitly supporting visual cue–oriented reasoning. By combining free-form reasoning with visual-cue rethinking, it mitigates failures from misleading or missing visual inputs and yields robust, adaptable performance across diverse multimodal scenarios.

## 3.2 PATTERN-GUIDED COLD START

While RLVR can steer models toward desired behaviors, relying on it alone is challenging and computationally costly—especially when the visual rethinking routine is relatively new to the model. To address this, we propose a Pattern-Guided Cold Start stage with Judge-Guided Seletive Training, initialized with a carefully curated 7K training set containing annotated reasoning trajectories across multiple domains, constructed via our iterative multimodal annotation pipeline, as shown in Figure 4.

**Iterative Multimodal Annotation Pipeline.** Several method have tried to construct reasoning dataset to improve multimodal reasoning. Heuristic-driven approaches (Xu et al., 2024) often lack the diversity to cover complex tasks, while methods (Huang et al., 2025) generating reasoning from

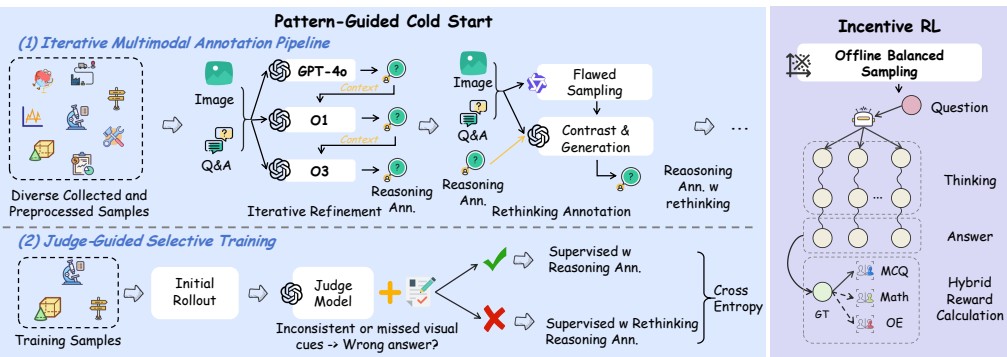

Figure 4: Overview of GThinker Training Pipeline. We first leverage the Iterative Multimodal Annotation Pipeline to generate high-quality reasoning data w. and w.o. visual thinking following the Cue-rethinking Pattern. Building on the constructed data, Judge-Guided Selective Training within the Pattern-Guided Cold Start phase then teaches the model how to think with the proposed pattern and when to rethink. Finally, Incentive Reinforcement Learning with DAPO enhances GThinker's ability to perform adaptive and accurate multimodal reasoning across diverse scenarios.

visual caption tend to produce text-centric deduction and suffers from inadequate visual information from captions. To address these shortcomings, we propose the Iterative Multimodal Annotation Pipeline to construct a dataset that unifies robust textual reasoning with solid visual cues for general reasoning, as shown in Figure 3. Instead of relying solely on text, we feed the image, question, and answer into an advanced multimodal model and prompt it to generate a reasoning process incorporate critical marked visual interpretation and deduction, as well as human-like cognitive processes such as self-reflection, ensured by the well-craft prompt. Next, we employ iterative refinement to mitigate incomplete logic or missed visual cues, which may occur when using a single model. The output from former model serves as context for a successor model, which is then prompted to correct, refine, and supplement the prior reasoning. Furthermore, to generate annotations that include visual rethinking, we devise a "flawed sampling-contrast-generation" process. To avoid the hallucinations of manual creation, we first collect diverse, incorrect reasoning samples using high-temperature roll-outs. Powerful O3 is then tasked with contrasting these flawed examples against our correct, refined annotations to generate new samples incorporating a cue-rethinking procedure. After automated verification as detailed in A., this pipeline yielded 7,358 samples across various domains, each labeled to indicate the presence or absence of a visual re-thinking process. We detailed data distribution, pre-processing and post-processing procedure, and prompts used in each step in the Appendix A.

**Judge-Guided Selective Training.** While reflection and re-thinking can significantly improve model accuracy, mandating this process for all training samples is suboptimal, and Naive approaches, such as randomly selecting a subset of samples for rethinking (Yao et al., 2024; Peng et al., 2025), also ignore critical differences among samples and model capabilities. Beyond teaching a model how to re-think, instilling the knowledge of when to trigger this process is equally crucial, especially from a cold start. To address this, we introduce a training strategy termed Judge-Guided Selective Training. We begin with an initial inference pass, rolling out the base model on the training set. Leveraging the LLM-as-a-judge paradigm (Gu et al., 2024), we then feed the question, answer, and the model's generated response to the employed judge model GPT-4o to diagnose the model's response to determine if errors are rooted in flawed visual cues. Based on the diagnostic step, we apply supervision using the detailed annotations that include the cue rethinking process for the samples where the model failed due to visual reasoning errors. Crucially, unlike rejection sampling, which discards incorrect responses, our method transforms these specific failures into valuable learning signals. This approach guides the model to learn not only how to reason following the cue-rethinking pattern but, more importantly, to recognize the very conditions that necessitate it.

### 3.3 INCENTIVE REINFORCEMENT LEARNING

Building on this foundation of Pattern-Guided Cold Start phase, we further enhance the model using reinforcement learning with verified rewards to encourage exploration and help it generalize across

diverse tasks and scenarios. Given recent advances in RLVR, we adopt the Decoupled Clip and Dynamic Sampling Policy Optimization (DAPO) algorithm (Yu et al., 2025) with a hybrid reward design supported by the offline balancing sampling.

**Preliminaries about DAPO.** DAPO improves from the GRPO (Shao et al., 2024b) by tackling two fundamental challenges in training for complex reasoning, as shown in Equation 1. First, it ensures training stability and efficiency through a Clip-Higher strategy and Dynamic Sampling. Second, it refines the supervision signal for long-chain reasoning by integrating Token-Level Policy Gradient Loss and Overlong Reward Shaping. These components enable fine-grained reward assignment and mitigate noise from reward model biases against verbosity, guiding the model to master high-quality reasoning patterns. Therefore, the model is better equipped to sample diverse reasoning paths and is enabled to learn reasoning strategies such as reflective knowledge inference for math tasks or cue-based rethinking in general multimodal scenarios. As a result, the model improves the ability to dynamically select the most suitable reasoning strategy for each situation, improving both generalization and robustness across domains.

$$\mathcal{J}_{\text{DAPO}}(\theta) = \mathbb{E}_{(q,a) \sim \mathcal{D}, \{o_i\}_{i=1}^G \sim \pi_{\text{old}}(\cdot|q)}$$

$$\left[ \frac{1}{\sum_{i=1}^G |o_i|} \sum_{i=1}^G \sum_{t=1}^{|o_i|} \min \left( r_{i,t}(\theta) \hat{A}_{i,t}, \text{clip} \left( r_{i,t}(\theta), 1 - \varepsilon_{\text{low}}, 1 + \varepsilon_{\text{high}} \right) \hat{A}_{i,t} \right) \right] \quad (1)$$

$$\text{s.t.} \quad 0 < |\{o_i \mid \texttt{is\_equivalent}(a, o_i)\}| < G,$$

where

$$r_{i,t}(\theta) = \frac{\pi_\theta(o_{i,t} \mid q, o_{i,<t})}{\pi_{\theta_{\text{old}}}(o_{i,t} \mid q, o_{i,<t})}, \hat{A}_{i,t} = \frac{R_i - \text{mean}(\{R_i\}_{i=1}^G)}{\text{std}(\{R_i\}_{i=1}^G)}. \quad (2)$$

**Hybrid Reward Design.** The default DAPO setting combines format-based and accuracy-based rewards. Prior approaches often constrain QA tasks to rigid formats, such as multiple-choice, and depend on exact string matching to assess correctness. This limits the range of question types the model can handle, especially in general scenarios, and model-based verification further reduces training efficiency. To overcome these limitations, we propose a hybrid reward strategy within the constraints of verifiable rewards. We support three main question types: multiple-choice, math, and simple open-ended formats. For multiple-choice questions, we apply exact answer matching. For math problems—whether numeric or symbolic—we use Math-Verify (Hynek, 2023) to extract and verify answers. For open-ended questions, we guide the model to summarize the answer in a standardized, concise format like a word or short phrase, enabling straightforward matching during reward computation. This design expands the diversity of supported question types while preserving reward accuracy. For the format reward, we follow prior work by enforcing and verifying adherence to the think-answer structure.

**Offline Balanced Sampling.** To support the reinforcement learning stage, we follow the practice (Yang et al., 2025; Xu et al., 2024) to collect diverse samples spanning math, science, and general reasoning tasks. However, we observed that not all samples are equally beneficial for training, given their varying types and difficulty levels. Therefore, to construct a balanced and effective dataset, we perform an offline balancing sampling procedure prior to the RL training. We first extract the joint embedding of image and question, and then follow (Vo et al., 2024) to perform clustering and sampling on these embeddings. Unlike heuristic approaches that rely on fine-grained manual categorization, our method focuses more on the image and question themselves. Subsequently, we conduct a rollout (n=16) on the remaining samples and discard instances where the model consistently fails to produce a correct response, which likely represent either annotation errors in the public source data or are prohibitively difficult. This curation pipeline yields a final set of 4K high-quality samples for the RL stage. We provide further details on the data composition in Appendix A.

## 4 EXPERIMENTS

### 4.1 IMPLEMENTATION DETAILS

**Training Settings.** We implement GThinker with the advanced MLLM Qwen2.5-VL-7B (Bai et al., 2025), one of the latest and most capable models at this scale, combining strong visual understanding

Table 1: Main results on comprehensive multimodal reasoning benchmark M³CoT. Abbreviations used in the table: Lang. (Language), Nat. (Natural), Soc. (Social), Phys. (Physical), Temp. (Temporal), Alg. (Algebra), Geom. (Geometry), Theo. (Theory). Excluding closed-source models, values in bold represent the highest performance, while underlined values indicate the second-best performance across all models.

| Model | Science | | | Commonsense | | | Mathematics | | | Overall |
|---|---|---|---|---|---|---|---|---|---|---|
| | Lang. | Nat. | Soc. | Phys. | Soc. | Temp. | Alg. | Geom. | Theo. | |
| *Closed-Source Models* | | | | | | | | | | |
| Gemini-2.5 Pro (DeepMind, 2025) | 97.6 | 91.6 | 75.3 | 92.2 | 81.4 | 94.3 | 81.1 | 78.8 | 61.9 | 85.9 |
| O3-20250416 (OpenAI, 2025) | 96.2 | 89.3 | 68.0 | 91.1 | 80.2 | 93.5 | 95.0 | 87.5 | 90.5 | 83.8 |
| O4-mini-20250416 (OpenAI, 2025) | 97.2 | 84.7 | 62.9 | 94.4 | 82.6 | 91.1 | 92.9 | 86.3 | 76.2 | 80.9 |
| GPT-4o-20241120 (Hurst et al., 2024) | 96.7 | 72.0 | 58.3 | 91.1 | 76.4 | 82.9 | 21.4 | 31.3 | 23.8 | 67.4 |
| *Open-Source Models* | | | | | | | | | | |
| InternVL-2.5-8B (Chen et al., 2024b) | 82.5 | 63.7 | 45.2 | 86.7 | 79.8 | 93.4 | 42.8 | 27.5 | 33.3 | 61.8 |
| Ovis2-8B (Lu et al., 2024) | 80.6 | 63.1 | 46.2 | 83.3 | 79.3 | 87.8 | 45.0 | 42.5 | 38.9 | 61.9 |
| Valley2(Wu et al., 2025) | 85.3 | 64.4 | 48.4 | 90.0 | 77.7 | 80.5 | 43.6 | 36.3 | 47.6 | 62.8 |
| Qwen2.5-VL-7B (Bai et al., 2025) | 82.9 | 61.2 | 46.8 | 82.2 | 81.4 | 81.3 | 57.9 | 40.0 | 61.9 | 62.4 |
| *Reasoning Models* | | | | | | | | | | |
| LLaVA-CoT-11B (Xu et al., 2024) | 72.0 | 56.4 | 41.7 | 84.4 | 72.3 | 82.1 | 37.9 | 36.3 | 33.3 | 56.0 |
| InternVL2.5-MPO-8B (Wang et al., 2024b) | 92.4 | 75.9 | 61.9 | 85.6 | 82.6 | 94.3 | 55.0 | 43.8 | 76.2 | 73.3 |
| Kimi-VL-A3B-Thinking (Team et al., 2025) | 86.2 | 64.4 | 39.6 | **91.1** | 78.9 | 89.4 | 13.5 | 15.0 | 14.2 | 58.3 |
| MM-Eureka-7B (Meng et al., 2025) | 86.7 | 71.5 | 57.3 | 81.1 | 80.2 | 90.2 | 40.0 | 23.8 | 28.6 | 67.4 |
| R1-OneVision-7B (Yang et al., 2025) | 74.9 | 66.4 | 51.4 | 84.4 | 72.3 | 85.4 | 30.0 | 31.3 | 42.9 | 61.8 |
| VLAA-Thinker-7B (Chen et al., 2025a) | 91.0 | 70.6 | 58.1 | 78.9 | 78.1 | 87.8 | 45.7 | 35.3 | 28.6 | 68.0 |
| GThinker-7B | **92.4** | **90.7** | **68.9** | 82.2 | **81.4** | **94.3** | **73.5** | **62.5** | **81.0** | **81.5** |

with broad general knowledge. We train the GThinker using our design two-stage pipeline, including pattern-guided cold start and incentive reinforcement learning with the constructed data. For Pattern-Guided Cold Start, we use a global batch size of 128 and a learning rate of 5e-6, training the model with the 7K reasoning path annotated data for 3 epochs. In the Incentive RL stage, we set the rollout number to 16, use a global batch size of 64, and start with a learning rate of 1e-6, training for 170 steps using the curated 4K data. Training is conducted on 4 nodes, each with 8 NVIDIA H100 GPUs. The total training time is about 9 hours. We provide more details in Appendix E.

**Evaluation Settings.** We evaluate our model against leading closed-source (e.g., O4-mini) and open-source structured and RL reasoning models on a diverse suite of benchmarks. We mainly include M³CoT, a challenging benchmark requiring multi-step reasoning based on visual cues across science, commonsense, and math. To provide a more fine-grained analysis, we also evaluate performance on several domain-specific benchmarks, including general reasoning, scientific reasoning, and mathematical Reasoning. A detailed description of the benchmarks and evaluation setting can be found in the Appendix C.

## 4.2 MAIN RESULTS

In the main results, we mainly compare our GThinker-7B based on Qwen2.5-VL-7B with leading reasoning models, and provide more results with larger 32B model and different baseline models in the Appendix B.

**Comparison with Baseline and Structured Reasoner.** GThinker-7B significantly surpasses both strong base models and structured long-chain reasoners like LLaVA-COT without carefully pre-defined reasoning steps. As shown in Table 1, our model achieves an impressive 81.5% on the comprehensive and challenging M³CoT benchmark, with a substantial +19.1 improvement over the strong baseline Qwen-VL-2.5-7B and an even more commanding +25.5 lead over the LLaVA-CoT-11B model. This advantage further proves our model's advanced ability to integrate visual and logical steps flexibly. Meanwhile, such superiority is further validated on specialized benchmarks as shown in Table 2. GThinker-7B consistently outperforms both Qwen-VL-2.5 and LLaVA-CoT across all evaluated tasks, with an average improvement of +2.1 on general reasoning MMStar and RealWorldQA, +2.4 on multidisciplinary MMMU-Pro, and +4.5 on MathVista.

Table 2: Main results on math-related and multidisciplinary benchmarks, and also fine-grained understanding of multimodal benchmarks incorporating reasoning. We use the setting detailed in the evaluation settings.

| Model | MMStar | RealWorldQA | MMMU-Pro | MathVista | MathVision |
|---|---|---|---|---|---|
| *Close-Source Models* | | | | | |
| Gemini-2.5 Pro | 73.6 | 78.0 | 68.8 | 80.9 | 73.3 |
| GPT-4o-20241120 | 65.1 | 76.2 | 54.5 | 63.8 | 31.2 |
| *Open-Source Models* | | | | | |
| InternVL2.5-8B (Chen et al., 2024b) | 62.8 | **70.1** | 34.4 | 64.4 | 19.7 |
| Ovis2-8B (Lu et al., 2024) | 64.4 | - | - | 71.4 | 25.9 |
| Valley2 (Wu et al., 2025) | 62.5 | 67.5 | - | 69.1 | 24.9 |
| Qwen2.5-VL-7B (Bai et al., 2025) | 63.9 | 68.5 | 38.3 | 68.2 | 25.1 |
| *Reasoning Models* | | | | | |
| LLaVA-CoT-11B (Xu et al., 2024) | 57.6 | 63.6 | 33.8 | 54.8 | 20.6 |
| InternVL2.5-MPO-8B (Wang et al., 2024b) | - | - | - | 67.0 | 25.7 |
| Kimi-VL-A3B-Thinking (Team et al., 2025) | 60.8 | - | - | 67.6 | **36.8** |
| MM-Eureka-7B (Meng et al., 2025) | 64.2 | 67.3 | **40.7** | **73.0** | 26.9 |
| R1-Onevision-7B (Yang et al., 2025) | 42.8 | 62.7 | 31.0 | 64.1 | 29.9 |
| VLAA-Thinker-7B (Chen et al., 2025a) | 63.7 | 66.9 | 39.8 | 68.0 | 26.4 |
| GThinker-7B | **66.4** | **70.1** | **40.7** | 72.7 | 26.6 |

**Comparison with RL Reasoners.** We further compare our method with leading RL reasoning methods, including both the MPO method InternVL2.5-MPO-8B and the RLVR methods with different training recipes. In summary, GThinker-7B establishes itself as not only the leading performer but also the most versatile. As shown in Table 1, on challenging M³CoT, GThinker-7B's score of 81.5% is the highest among all open-source models, outperforming the previous SOTA InternVL2.5-MPO-8B and also competitive RL reasoners, including VLAA-Thinker-7B by +13.5, MM-Eureka-7B by +14.1, and Kimi-VL-A3B-Thinking by +8.2. Crucially, GThinker-7B avoids the common performance trade-off observed in RL-trained reasoners. As we claimed as an asymmetry, RL can enhance MLLMs' long-chain reasoning ability on mathematical tasks, but degrades on general and multidisciplinary benchmarks, based on previous studies. For example, as seen in Table 2, both VLAA-Thinker-7B and MM-Eureka-7B underperform their baselines on general benchmarks like MMStar. In contrast, GThinker-7B achieves 72.7% on MathVista (+4.5 points over baseline) and 26.6% on MathVision (+1.5 points). Similarly, on the multidisciplinary science benchmark MMMU-Pro, GThinker-7B improves by approximately 4 points. Furthermore, it shows significant gains on general benchmarks requiring fine-grained understanding and further reasoning, with 66.4% on MMStar and 70.1% on RealWorldQA. This demonstrates our method's unique effectiveness in fostering a general reasoning capability by unifying the textual reasoning with visual cue rethinking.

### 4.3 ABLATION STUDY

**Ablation on the Iterative Multimodal Reasoning Pipeline.** High-quality data is crucial for training effective multimodal reasoning models. To build a high-quality multimodal reasoning data aligned with the proposed cue-rethinking pattern, we propose the iterative multimodal annotation pipeline. We compare our method with the text input only (caption + question) data construction method (Huang et al., 2025) indicated by Caption in Table 3 under the same generation model to validate the necessity and advantage of multimodal annotation, as well as the iterative refinement strategy we use. As shown in Table 3, using only the text

Table 3: Ablation on the Iterative Multimodal Reasoning Pipeline. Iter. indicates iterative refinement.

| Caption | Image | Iter. | M³CoT |
|---|---|---|---|
| ✓ | | | 63.5 |
| | ✓ | | 69.6 |
| | ✓ | ✓ | 73.6 |

inputs with caption replacing the image, yields an overall score of 63.5%. Our data generation pipeline, even without iterative refinement, significantly improves performance to 69.6% (+6.1%

Table 4: Ablation on GThinker Components.

| Method | Science | Com. | Math | Overall |
|---|---|---|---|---|
| Qwen2.5-VL-7B | 57.6 | 80.8 | 60.6 | 62.4 |
| Qwen2.5-VL-7B-Zero | 63.3 | 81.6 | 49.0 | 64.2 |
| + Patten Guided Cold Start | 73.1 | 79.3 | 46.9 | 73.6 |
| *w/o* Judge-Guided Selective Training | 68.0 | 82.0 | 42.7 | 68.4 |
| + Incentive Reinforcement Learning | 82.5 | 83.7 | 71.0 | 81.5 |
| *w/o* Offline Balanced Sampling | 82.2 | 84.2 | 60.2 | 80.4 |

absolute). Incorporating our iterative refinement process further boosts the overall score to 73.6%, an additional 4.0% improvement. We attribute this gain to the complementary strengths of the leading models, including GPT-4o, O1, and O3: *during the collaborative annotation iterations, visual cues and reasoning logic are more thoroughly captured, further boosting the quality of the CoT data.*

**Ablation on GThinker Components.** As shown in Table 4, we conduct ablation studies to examine the individual contribution of each design in GThinker, evaluating on M³CoT, including the pattern-guided cold start with Judge-Guided Selective Training and Incentive Reinforcement Learning with Offline Balanced Sampling. Compared with the baseline, incorporating the pattern-guided cold start yields a performance boost of +11.2%, with Judge-Guided Selective Training contributing 5.2% improvement. Such a result highlights the effectiveness of our Judge-Guided Selective Training to learn from visual cue failure cases to learn when to rethink. Further training the model with Incentive Reinforcement Learning brings an improvement of 6.9%, among which Offline Balance Sampling contributes 1.1%. Meanwhile, compared with the model solely training with the DAPO, termed Qwen2.5-VL-7B-Zero, we lead by a large margin. These results verify that such visual cue-rethinking is different from text-centric reflection, which cannot be reliably induced by verifiable-reward RL only, and demonstrate the effectiveness of our cue-rethinking pattern combined with the two-stage training recipe.

## 5 QUALITATIVE ANALYSIS

To validate our Cue-Rethinking pattern in practice, we qualitatively analyze the model's generation process, presenting two representative case studies in the Appendix Figure 6 and Figure 7. As shown in the Figure 7, GThinker can augment and revise visual cues during the reasoning phase, ultimately leading to the correct solution, when essential. With correct and adequate visual cues, GThinker can also critically reflect upon and validate its reasoning pathway from both logical and computational standpoints to ascertain the final answer for math problems in Figure 6. These instances effectively highlight the adaptability of our designed pattern to diverse problems and tasks by accommodating varied thinking approaches, thereby underscoring the success of our training regimen.

## 6 CONCLUSION

In this paper, we identify a fundamental limitation in multimodal reasoning: the inability to perform adaptive visual rethinking, a process crucial for robust reasoning in general-purpose scenarios. We argue that this capability cannot be reliably induced by reinforcement learning from visual feedback alone, nor can it be captured by rigid, template-based reasoning structures. It requires a new mechanism to guide rethinking, making it as natural and evidence-grounded as textual reasoning. To address this, we introduce GThinker, a framework that learns a novel reasoning pattern called the Cue-Rethinking Pattern. This pattern compels the model to eschew fixed formats and instead ground its reasoning and rethinking in visual cues. Through a two-stage training of pattern-guided cold start and incentive RL, GThinker effectively unifies robust textual reasoning, such as reflection, with essential visual rethinking. Extensive experiments on multi-domain multimodal reasoning benchmarks show that GThinker outperforms existing reasoning MLLMs in both accuracy and cross-domain adaptability. Ablation studies further confirm the effectiveness of each core design component.

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
