# A GTHINKER-11K CONSTRUCTION

To support the training of GThinker, we have designed a scalable data generation pipeline to construct the GThinker-11K data as we have concluded in §3.2 and §3.3, respectively. In this section, we systematically introduce the data construction process, including the 7K cold start data, as depicted in Figure 5, and 4K RL data.

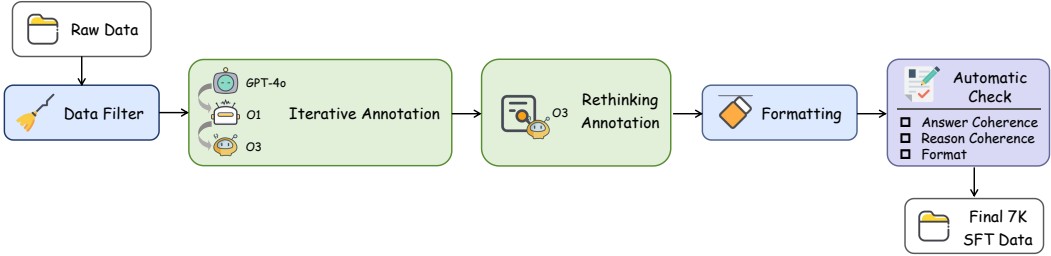

Figure 5: Full Iterative Multimodal Annotation Pipeline.

## A.1 DATA PREPARATIONS

Though several datasets are constructed to enhance multimodal reasoning capabilities in MLLMs (Yao et al., 2024; Xu et al., 2024; Yang et al., 2025) spanning diverse domains, they often present challenges such as high knowledge dependency, limited visual cues, or limited reasoning level. To extend the multimodal reasoning to general scenarios beyond knowledge-intensive math and science problems, we empirically find that the $M^3CoT$ dataset provides a well-established data baseline for multimodal reasoning across domains. It details how to collect data across science, mathematics, and general scenarios with commonsense, and ensure the visual reliance and reasoning complexity with final manual checking. Building on baseline, we apply a two-step filtering process to ensure data quality: (1) we discard entries with corrupted or missing images, and (2) we verify the remaining samples' compliance with closed-source model usage policies using GPT-4o, resulting in 7,358 high-quality samples. We illustrated the data composition in Table 5.

Table 5: Data composition of 7K Cold Start data of GThinker-11K.

| Type | Volume | Source |
|---|---|---|
| Science | 5266 | KiloGram(Ji et al., 2022), ScienceQA (Lu et al., 2022a), $M^3CoT$ (Chen et al., 2024a) |
| Mathamatics | 621 | TableWMP (Lu et al., 2022b), Math (Hendrycks et al., 2021) |
| Commonsense | 1471 | Sherlock (Hessel et al., 2022)(Questions generated by $M^3CoT$) |

## A.2 MULTIMODAL ITERATIVE ANNOTATION

To generate high-quality reasoning paths and visual cues, we propose a multimodal iterative annotation methodology that leverages multiple leading MLLMs, such as OpenAI's O-series, for end-to-end reasoning path generation, different from prior approaches (Wu & Xie, 2024; Yang et al., 2025; Yao et al., 2024) that rely on multi-step pipelines which generate captions first and then utilize the reasoning LLMs. This leads to more efficient generation and results in more coherent multi-modal long-chain reasoning paths, richer step-by-step visual cues, and stronger logical deductions. As shown in Figure 5, drawing on the insight that different models offer complementary strengths (Yao et al., 2024), we implement an iterative refinement strategy: initial annotations from Qwen2.5-VL-7B, as models with lower parameters sometimes are more faithful to the visual content, and is first revised by GPT-4o to reduce apparent errors. Then, the results are processed by O1 and further enhanced by O3. To finish this, we guide the models using carefully engineered prompts optimized through few-shot learning, as shown in **Prompt 1**. For each image–question–answer triplet,

the model is prompted to produce a long reasoning process or refine the long reasoning chain with the relevant visual cues identified. This three-stage process significantly improves the accuracy and depth of final thinking annotations by leveraging the diverse capabilities of each model.

### A.3 RETHINKING ANNOTATION

With the positive, high-quality reasoning data, we further extend our process to handle negative reasoning with corrections. Rather than manually crafting incorrect reasoning traces (Zhan et al., 2024; Zhang et al.), which may introduce artifacts due to the gap between human-designed prompts and model capabilities, we first sample natural, flawed responses from 7B-level capable but compact models (Bai et al., 2025; Wu et al., 2025). While positive samples provide a reference point for correction, the variability in natural language expression requires a more nuanced approach. To this end, we employ the advanced reasoning capabilities of O3. Using carefully designed prompts, as shown in **Prompt 2**, we guide the model to compare incorrect reasoning against the correct reasoning path and the corresponding image. This enables the model to identify and correct missing or uncertain and misleading visual cues and faulty inferences. For visual cue correction, each initial cue is explicitly linked to its corrected counterpart, followed by the revised deduction, ensuring the data remains structured and easy to parse.

### A.4 FORMATTING

After all annotations are completed, we utilize GPT-4o to parse and format all the data. This includes standardizing elements like line breaks within the <think></think><answer></answer>format and extracting the correct, key visual cues. This process is designed to facilitate broader subsequent use.

### A.5 AUTOMATIC VERIFICATION

With the formatted annotated data, we perform automatic checks targeted at three critical aspects to ensure high data quality, helped by annotation-excluded Gemini 2.5 Pro (DeepMind, 2025), as illustrated in Figure 5. These checks target three critical aspects. First, for format validation, we ensure that for each annotation, the positive reasoning path ends with a concluded answer, and the visual cues can be parsed. Second, for answer consistency, the annotated answers are parsed and cross-checked against the ground truth. Third, for reasoning coherence, we input the image, QA pair, and annotated reasoning into Gemini 2.5 Pro to evaluate logical alignment between visual cues and reasoning with **Prompt 3**, flagging any contradictions. Samples with identified issues are reprocessed through the relevant correction steps in our pipeline. Samples with identified issues are reprocessed through the relevant correction steps in our pipeline.

To assess the quality control of the designed pipeline, we manually review a randomly selected 15% subset of the final dataset and confirm that our pipeline reliably produces high-quality annotations, which ensures scalability.

### A.6 REINFORCEMENT LEARNING DATA CONSTRUCTION

We first collect data from a broader range of sources (Meng et al., 2025; Xu et al., 2024; Yang et al., 2025) to ensure the generalization to different scenarios encompassing the general scenarios, math, and science. Instead of directly employing these data, we adopt the proposed offline balanced sampling methodology from (Vo et al., 2024) to cluster and curate 4K samples. We illustrate the composition of the final 4K data in Table 6.

Table 6: RL data composition.

| Type | Volume |
|---|---|
| Mathematics | 748 |
| Science | 1557 |
| General | 1719 |

## B EXPERIMENTS ON MORE BASELINE MODELS

To rigorously evaluate the generalizability of GThinker, we conduct experiments on the latest 7B leading models, Valley-7B and Ovis-7B, on the closed-source opencompass leaderboard, and also

on a larger scale Qwen2.5-VL-32B. We evaluate their models under the same setting. As shown in Table 7, GThinker yields consistent gains across these diverse baselines, confirming its broad applicability. Meanwhile, GThinker improves the performance of these baselines on the general Q&A dominant Leaderboard, OpenCompass Academic. These promising results further confirm its broad applicability.

Table 7: Experiments on scaled and other baseline models

| Model | M$^3$CoT | MMStar | MMMU_Pro | MathVista |
|---|---|---|---|---|
| Valley2-7B | 63.5 | 59.5 | 31.2 | 63.2 |
| GThinker-Valley2-7B | 74.8 | 62.1 | 38.7 | 65.5 |
| Ovis2-8B | 62.9 | 62.9 | 36.8 | 69.5 |
| GThinker-Ovis2-8B | 74.2 | 63.5 | 37.4 | 72.4 |
| Qwen2.5-VL-32B | 72.7 | 68.8 | 44.0 | 72.9 |
| GThinker-Qwen2.5-VL-32B | 81.9 | 69.6 | 50.8 | 73.9 |

## C  EVALUATION SETTINGS

All evaluations are conducted on a single node equipped with 8 NVIDIA H100 GPUs. For M$^3$CoT, we follow each model's official settings and prompts and use VLMEvalKit (Duan et al., 2024) for fair evaluation. For other benchmarks, we use the results reported in their original papers. For RL-enhanced reasoning models, which primarily focus on math and science domains, we follow their released models and evaluation guidelines to conduct testing. The evaluation focuses on multimodal reasoning across general, mathematical, and scientific scenarios:

- **M$^3$CoT:** A challenging benchmark that spans science, commonsense, and math domains, with each example verified to require multi-step reasoning. We primarily use this benchmark to comprehensively evaluate models' multimodal reasoning capabilities across diverse scenarios.

- **General scenario benchmarks:** MMStar (Chen et al.) and RealWorldQA (xAI, 2024). These benchmarks focus on general and realistic scenarios, including parts of understanding-based reasoning tasks, and are used to evaluate multimodal reasoning capabilities.

- **Science and math scenario benchmarks:** We use MMMU-Pro (Yue et al., 2024), which covers multiple scientific subjects, to evaluate multimodal reasoning in scientific contexts. For math-specific evaluation, we adopt the widely used MathVista (Lu et al.) and MathVision (Wang et al., 2024a)benchmarks.

## D  QUALITATIVE ANALYSIS

This section presents more examples to showcase the efficacy of our proposed method. As illustrated in Figure 6, GThinker, subsequent to our training, demonstrates the ability to augment and revise visual cues during the reasoning phase, ultimately leading to the correct solution. As we demonstrated in §3.1, such re-evaluation of visual cues is not invariably essential. Therefore, for multimodal reasoning tasks, including mathematics, our pattern supports that once adequate visual information is assimilated, the model can engage in direct reasoning flexibly with critical reflection and verification. As depicted in Figure 7, GThinker can also critically reflect upon and validate its reasoning pathway from both logical and computational standpoints to ascertain the final answer for math problems with accurate visual cues identified. These instances effectively highlight the adaptability of our Cue-Rethinking Pattern to diverse problems and tasks by accommodating varied thinking approaches, thereby underscoring the success of our training regimen.

**Prompt 1: Multimodal Iterative Annotation Prompt**

You are a Checker-&-Corrector-&-Annotator of multimodal chain-of-thought answers.

**Input you will receive (always in this order)**
1. The multi-choice question with the corresponding image.
2. The true answer label (e.g. "B").
3. A short, human-annotated rationale for that true answer.
4. The model's PREVIOUS reasoning response, formatted exactly as

<think> ...model's chain-of-thought (CoT)... </think>
<answer> ...model's final letter or text answer... </answer>

• Inside the <think>...</think> block, visual cues that the model claims to use are wrapped as <vcues_1> ... </vcues_1>, <vcues_2> ... </vcues_2>, etc.

**Your task:**
A. Verify the correctness of the previous model's answer and reasoning against the given image, true answer and human rationale.
B. If the model's final answer is already correct, keep the answer part.
C. If the answer is correct but some visual cues or reasoning steps are wrong or missing, fix the wrong cues / steps and append the NECESSARY cues/steps according to your knowledge.
D. If the answer is wrong, repair the erroneous cues / logic so that the corrected reasoning leads to the true answer.
E. Preserve structure, ordering and tags as possible—modify ONLY what is necessary for correctness and clarity.
F. Keep all tag syntax unchanged (<think> ... </think>, <answer> ... </answer>, <vcues_*> ... </vcues_*>) so the output can be parsed automatically.

**Output format**
Return ONE corrected response, nothing else, in exactly the same two-tag layout:

<think>
...corrected chain-of-thought with fixed <vcues_*></vcues_*>...
</think>
<answer>
...single correct choice or textual answer...
</answer>

**Additional rules**
• If you remove an incorrect visual cue, replace it with the correct cue and keep the numbering consistent.
• Never fabricate content outside the scope of the provided information.
• Be concise—do not add redundant and repeated explanations beyond what is needed for a logically sound, correct solution.

**Examples**
• *Example 1*
• *Example 2*

> **Prompt 2: Rethinking Annotation Prompt**
>
> You are a Visual Reasoning Corrector and Annotator. Process the input ¡Model_Infer¿ with these rules:
>
> 1. **Response Segmentation**:
> - Remove the answer conclusion part in the model.
> - Then, wrap the model's entire thought process in <think></think>.
>
> 2. **Visual Cues Annotation**:
> - Within the <think> section, identify specific visual cue phrases (not entire paragraphs) and annotate each one with a tag in the format <vcues_*></vcues_*>, starting numbering from 1 (i.e. <vcues_1>, <vcues_2>, ...).
>
> 3. **Visual Cues Reasoning Error Diagnosis and Correction**:
> 3.0. All the data to be processed now concern reasoning errors based on visual cues and may also include errors in visual cues. These reasoning errors may include issues such as insufficient knowledge, over-analysis, etc.
> 3.1. **During this process, do not revise the model's previous entire original thought after annotation**
> 3.2. Before the closing </think> tag, and insert a generated transitional sentence wrapped with <aha></aha> that conveys a message similar in meaning to: "Let's check each visual cue and corresponding reasoning before giving the final answer. Generate the error type based on the Error Pre-judgement: It looks like the visual cues are correct with some reasoning error." (The exact wording can vary as long as the idea is the same.)
> 3.3. On the next line immediately after this transitional sentence, for each visual cue annotated (using <vcues_*></vcues_*>) and their corresponding reasoning parts before <aha>, compare them with :
> - The verified rationale (<rationale>)
> - Your understanding of image
> Then, after </aha>, update the corrected reasoning based on the visual cues. If necessary, replicate the relevant part from the original <vcues_*></vcues_*> tag alongside the revised reasoning.
> 3.4. After completing the reasoning corrections, perform a logical verification of the reasoning after the </aha> part
> 3.5. Append the final correct answer wrapped with <answer></answer>, i.e. <answer><Correct Answer></answer>, in the next line after the </think>, ensuring that the final answer is adjusted correctly.
>
> 4. **Output Constraints**:
> - Preserve the original reasoning structure as much as possible.
> - **Do not include similar phrases like "based on the rationale", "The reasoning should focus", "aligns with the rationale", "the model", because the processed content is used for the model training instead of a third-person view**
> - Ensure that all annotations (<think>, <answer>, <vcues_*>, <aha>) are properly formatted and inserted in the correct locations.
>
> Example 1:
> ...

> **Prompt 3: Verification Prompt**
>
> You are given a multiple-choice question with options and the image, the correct answer, and a generated response in the following format:
>
> <think>thinking process here</think>
> <answer>answer choice</answer>
>
> You should align the answer choice in <answer></answer> with the choice content in the question, and then check whether the reasoning in <think>...<think> logically supports the answer choice content.
>
> If the thinking process leads to that answer choice, output 1. Otherwise, output 0 and explain why it does not lead to the answer.

## E  TRAINING DETAILS

### E.1  SYSTEM PROMPT

For the training and evaluation of the GThinker, we utilize the same system prompt to wrap the conversation, as shown below.

> **System Prompt**
>
> A conversation between User and Assistant. The user asks a question, and the Assistant solves it. The assistant first thinks about the reasoning process in the mind and then provides the user with the answer. The reasoning process and answer are enclosed within <think> </think> and <answer> </answer> tags, respectively, i.e., <think> reasoning process here </think><answer> answer here </answer>. In the reasoning process enclosed within <think> </think>, each specific visual cue is enclosed within <vcues_*>...</vcues_*>, where * indicates the index of the specific cue. Before concluding the final answer, pause for a quick consistency check: verify whether the visual cues support the reasoning and whether each step logically follows from what is seen. If correct, conclude the answer; otherwise, revise the visual cues and reasoning, then conclude.

### E.2  HYPER-PARAMETERS

We have illustrated the key hyper-parameters in the §4.1. In this section, we provide more information about the hyper-parameters used in our experiment. For the DAPO, we utilize the EasyR1 framework for training.

Table 8: Hyper-parameters for Supervised Fine-tuning

| Name | Value |
| --- | --- |
| precision | bf16 |
| max_seq_length | 4096 |
| warmup_ratio | 0.1 |
| max_pixels | 12845056 |
| min_pixels | 316 |

Table 9: Hyper-parameters for DAPO

| Name | Value |
| --- | --- |
| max_promp_length | 15000 |
| max_response_length | 4096 |
| global_batch_size | 64 |
| rollout_batch_size | 64 |
| max_pixels | 4194304 |
| min_pixels | 262144 |
| weight_decay | 1e-2 |

## F  LLM USAGE

Our use of a Large Language Model (LLM) in this work was limited to providing word-level hints and occasional suggestions for sentence improvement. This is in accordance with the ICLR's Code of Conduct on LLM usage.

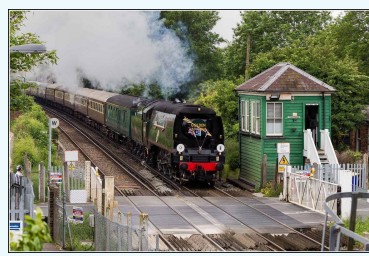

What is happening in the picture?
A. A man is riding his bike
B. The train has stopped at a small station
C. The train will pass by the gates
D. The train has broken down

<think>To answer this question, let's analyze the image step by step:

1. <vcues_1>The image shows a steam train on the tracks</vcues_1>.
2. <vcues_2>There is a small green building on the right side of the image, which appears to be a signal box or a small station</vcues_2>.
3. <vcues_3>The train is moving on the tracks, and there is no indication that it has stopped at a station</vcues_3>.
4. <vcues_4>There are no gates visible in the image, and the train is not shown breaking down</vcues_4>.

Let's verify each visual cue and its reasoning before finalizing the answer.

Some visual cues seem to need adjustment.

<vcues_2>The small green building on the right side of the image is a signal box, which is typically used to control train movements</vcues_2>. This suggests that the train is passing by a small station.

<vcues_4>There are gates visible in the image, and the train is on the tracks, which indicates that the train will pass by the gates</vcues_4>.

With the corrected visual cues, the most likely scenario is that the train is passing by the gates.</think>

<answer>C</answer>

Figure 6: Response sample from GThinker with visual cue-based rethinking enabled.

As shown in the figure, the straight line AD parallel BC, then the degree of angle 2 is () Choices: A:42°  B:50°  C:60°  D:68°

<think>To solve this problem, we need to use the properties of parallel lines and the angles formed by a transversal.

1. Identify the given information:
  - <vcues_1>AD is parallel to BC</vcues_1>.
  - <vcues_2>Angle 1 is 42°</vcues_2>.
  - <vcues_3>Angle 78° is an angle formed by the transversal and line AD</vcues_3>.

2. Use the properties of parallel lines:
  - Since AD is parallel to BC, the angle formed by the transversal and line AD will be related to the angle formed by the transversal and line BC.

3. Determine the relationship between the angles:
  - <vcues_4>Angle 2 is an exterior angle to the triangle formed by the transversal and lines AD and BC</vcues_4>.
  - The angle 78° is an interior angle of the triangle, and angle 1 is 42°.
  - The exterior angle (angle 2) is the sum of the two non-adjacent interior angles of the triangle.

4. Calculate the angle:
  - <vcues_5>Angle 2 = 78° - 42° = 36°</vcues_5>.

However, upon reevaluating the problem, it appears there might be a misunderstanding in the interpretation of the angles. The correct approach is to use the fact that angle 2 is the exterior angle to the triangle formed by the transversal and the lines AD and BC.

- <vcues_6>Angle 2 is the exterior angle to the triangle formed by the transversal and the lines AD and BC</vcues_6>.
- The exterior angle is equal to the sum of the two non-adjacent interior angles of the triangle.

Given the angles:
- <vcues_7>Angle 78° is an interior angle of the triangle</vcues_7>.
- <vcues_8>Angle 1 is 42°</vcues_8>.

The exterior angle (angle 2) is the sum of the two non-adjacent interior angles:
- <vcues_9>Angle 2 = 78° - 42° = 36°</vcues_9>.

However, upon rechecking the problem statement and the options, it appears there might be a mistake in the interpretation of the angles. The correct angle should be calculated as:
- <vcues_10>Angle 2 = 180° - 78° = 102° - 42° = 60°</vcues_10>.

Given the options, the correct answer is:
- <vcues_11>Angle 2 = 60°</vcues_11>.

Therefore, the correct choice is C.
Wait—cross-reference those cues with the logic.
The reasoning seems to hold up with what's been seen.</think>

<answer>C</answer>

Figure 7: Response sample from GThinker with reflection for the math problem.