# OpenReview forum: "GThinker: Towards General Multimodal Reasoning via Cue-Guided Rethinking"
_ICLR.cc/2026/Conference — ICLR 2026 Conference Withdrawn Submission_

### Official Review · Reviewer_mLSm · 2025-10-29

**Soundness:** 3
**Presentation:** 3
**Contribution:** 2
**Rating:** 6
**Confidence:** 4

**Summary:**

This paper proposes GThinker to address a core deficiency in MLLMs for visual reasoning: the lack of adaptive rethinking capability regarding visual cues. By introducing a Cue-Rethinking reasoning paradigm and a two-stage training process (pattern-guided cold start + incentive reinforcement learning), the GThinker significantly enhance the model's reasoning robustness in complex visual tasks. Experiments demonstrate that GThinker achieves SOTA performance on multimodal reasoning benchmarks such as M^3CoT and exhibits balanced advantages across general scenarios, mathematical reasoning and scientific reasoning tasks.

**Strengths:**

1. GThinker proposes a gap in MLLMs: uncritical acceptance of initial visual cues despite textual reflection capabilities. This provides a fresh perspective for multimodal reasoning research.

2. The Cue-Rethinking pattern innovatively combines free-form reasoning with visual cue grounding <vcues>, enabling flexible rethinking without rigid templates.

3. Iterative annotation pipeline generates high-quality data (GThinker-11k), mitigating hallucination through flawed sampling-contrast-generation. It has significantly improved performance on benchmarks such as M^3CoT, and in particular, it can enhance the general capabilities of MLLMs.

**Weaknesses:**

1. The claim that existing models “ignore key visual cues” lacks quantitative support. Empirical analysis (e.g., error-case statistics) is needed to validate this asymmetry.

2. Experiments are restricted to Qwen2.5-VL-7B. Validation on larger/alternative architectures can be explored to ensure generalizability.

3. Typo: Table 1 shows that InternVL2.5-MPO-8B is the best model on Commonsense Soc. (82.6).

**Questions:**

1. The hybrid reward design claims concise-format matching for open-ended answers, but no evidence is provided for its reliability. How are ambiguous answers (e.g., synonyms) handled? More details are needed to provide.

2. Figure 6 (Appendix) suggests GThinker generates verbose reasoning for simple tasks. This raises efficiency concerns. Does the Cue-Rethinking trigger mechanism adapt to task complexity?

3. Missing Baseline for Cue-Rethinking Prompt: A critical baseline is omitted: directly adding the Cue-Rethinking prompt (“Let’s check each visual cue…”) to existing models (e.g., InternVL2.5-MPO-8B) without training.

---

### Official Review · Reviewer_6irD · 2025-10-30

**Soundness:** 2
**Presentation:** 3
**Contribution:** 2
**Rating:** 4
**Confidence:** 5

**Summary:**

GThinker introduces a novel multimodal reasoning paradigm called Cue-Rethinking, addressing the issue of multimodal large language models lacking adaptive visual introspection. This method guides models to explicitly mark visual cues and strategically re-evaluate them during reasoning to resolve potential inconsistencies. To train this capability, a two-stage framework, comprising pattern-guided cold start and incentive reinforcement learning, was developed. A large-scale dataset, GThinker-11k, was constructed to support the model's learning of when and how to perform visual corrections. Empirical results demonstrate GThinker's substantial performance improvement over existing models on various comprehensive multimodal reasoning benchmarks. However, the approach has limitations regarding the generalizability of its training data, the infallibility of its judge models, and the direct mechanistic verification of its core "rethinking" process.

**Strengths:**

1. This paper excels in its conceptual introduction and methodological exposition. It clearly identifies a core asymmetry in multimodal reasoning and visually illustrates (e.g., in Figures 1, 2, and 3) the Cue-Rethinking pattern, its three-stage process, and the training pipeline in detail. This thorough and comprehensible presentation allows readers to quickly grasp GThinker's central innovative ideas and encourages further research.

2. GThinker's training framework demonstrates good reproducibility. The paper meticulously details the specific settings for both the Pattern-Guided Cold Start and Incentive Reinforcement Learning stages, including the foundational MLLM used (Qwen2.5-VL-7B), batch sizes, learning rates, training steps, and GPU configurations.

3. This paper clearly articulates the construction process of the GThinker-11k dataset, particularly outlining each step of the "Iterative Multimodal Annotation Pipeline." From initial reasoning generation and iterative refinement to the flawed sampling-contrast-generation, and the "Judge-Guided Selective Training" phase utilizing advanced models such as GPT-4o, O1, and O3 for diagnosis and correction.

**Weaknesses:**

1. The generalizability of the training data construction raises concerns. The study relies on samples extracted from and corrected based on the error patterns of a specific model (e.g., Qwen-VL) to guide the learning of "adaptive visual rethinking" behavior. This approach may lead to training data that is overly tailored to the specific deficiencies of the source model, thereby limiting the effective transferability of the GThinker strategy to other multimodal models (e.g., LLaVA) which possess distinct architectures and error profiles. Furthermore, if a target model inherently possesses correct reasoning capabilities for a given sample, but the training data, generated from another model's errors, imposes a need for "rethinking," this could compel the target model to learn redundant or inappropriate introspection processes, potentially compromising its inference efficiency and accuracy.

2. The reliability of the judge model as a "gold standard" constitutes an inherent weakness of this methodology. Although powerful, the advanced multimodal large language models (e.g., GPT-4o, O1, O3) utilized in this research to generate and refine the "visual rethinking" training data are not infallible. They may still commit errors in complex or ambiguous visual reasoning tasks, render misjudgments, or introduce their own biases (i.e., "hallucinations"), which are directly propagated into the generated training dataset. This implies a significant risk of error propagation: should the judge model's identification of visual cues, diagnosis of logical inconsistencies, or suggested correction paths be flawed, the GThinker model, in learning "when and how to rethink," may internalize these imperfections. Consequently, its acquired rethinking strategies might not be optimal, and could even be erroneous in certain scenarios.

3. The study lacks direct mechanistic evidence to unequivocally demonstrate that visual information is indeed "rethought" during the correction process. While the paper indirectly supports the efficacy of "rethinking" through modified textual reasoning chains and overall performance improvements, it does not provide underlying model interpretability evidence. For instance, there is an absence of direct visualization of attention maps correlating rethinking tokens with specific regions of the input image, nor is there a comparison of the model's attentional focus on critical visual cues "before" and "after" the rethinking phase. This reliance on indirect evidence diminishes the credibility of the claims regarding GThinker's core internal mechanisms and fails to conclusively prove that the model genuinely re-examines visual information at a fundamental processing level.

**Questions:**

1. The paper states GThinker is implemented based on Qwen2.5-VL-7B, and parts of the training data rely on Qwen-VL's error patterns corrected by GPT-4o/O1/O3. I would ask the authors if they have conducted further experiments to explore how the "rethinking" strategy learned through this method transfers to other foundational models with significantly different architectures and error profiles (ee.g., LLaVA series or InternVL models).

2. Given that judge models are not infallible and their potential errors or biases could be learned by GThinker, I would ask if the authors have performed sensitivity analyses. Specifically, have they evaluated the impact of using judge models of varying capabilities on GThinker's final performance and the quality of the learned "rethinking" strategies?

3. The paper indirectly supports the occurrence of "visual rethinking" through modified textual outputs. I would inquire whether the authors have plans or initial attempts to employ more direct, interpretable methods, such as analyzing changes in attention distribution between rethinking tokens and image regions before and after the rethinking phase, to provide more explicit evidence that the model indeed performs a meaningful re-examination of visual information internally.

---

### Official Review · Reviewer_KwUU · 2025-10-31

**Soundness:** 2
**Presentation:** 3
**Contribution:** 3
**Rating:** 2
**Confidence:** 4

**Summary:**

This paper propose a novel multimodal reasoning framework GThinker to address current reasoning mulitmodal LLM problems: (1) fixed reasoning (CoT) structure, limiting its domain to mathematics or science (2) they tend to uncritically accept initial visual interpretations and raly revise. The authors propose (1) cue-rethinking pattern marking explicitly visual cues in CoT traces, (2) pattern-guided cold start with judge-guided selective training that uses GPT-4o as judge model, even including the failed cases with additional annotation, and (3) reinforcement learning with verified rewards with hybrid reward design and offline balanced sampling to generalize across diverse domain for the model.

**Strengths:**

1. GThinker demonstrates superior SOTA performance on a wide variety of benchmarks, impressively generalizing across mathematics, science, and general-domain tasks
2. The "Judge-Guided Selective Training" is a novel training strategy. Using a judge to train selectively on failure cases (especially visual-based failures) is interesting.
3. The proposed GThinker-11k dataset is a valuable contribution to the community
4. nice and valuable ablation on the iterative data refinement

**Weaknesses:**

1. **Need for Clearer Substantiation of the Core Motivational Claims:** The paper's motivation hinges on two key assertions: (a) that general multimodal reasoning is more "reliant on visual interpretation" than math/science tasks, and (b) that MLLMs "uncritically accept initial visual interpretations" while being capable of correcting textual context. While these claims are intuitively plausible, the introduction presents them as foundational axioms without direct preliminary experiments or citations for support. To fully justify the specific need for "visual rethinking" (as opposed to general rethinking), the paper would be more persuasive if it first provided a brief analysis or cited evidence showing such behavior.
2. **Need for Clearer Differentiation from Prior "Rethinking" Work:** The paper's novelty rests on "Cue-Rethinking" being "fundamentally different" from prior text-centric reflection (e.g., [1]). The paper could be strengthened by providing a more direct analysis to support this claim. A key ambiguity is whether "Cue-Rethinking" primarily teaches the model to revise its initial visual interpretations (as suggested by Figure 2) or to perform a more general, text-centric re-evaluation of the existing cues. Without this clarification, the mechanism's novelty is less apparent, and the impressive gains might be (correspondingly) attributed more to the excellent training strategy (the judge-guided SFT) than to a fundamentally new reasoning pattern.
3. **Dependency on SOTA Models in the Training Pipeline**: The proposed training framework, particularly the iterative data pipeline and the "Judge-Guided Selective Training," shows a notable dependency on SOTA proprietary models like GPT-4o and O3. This raises practical concerns about reproducibility and cost for the wider community. Furthermore, it makes it difficult to fully disentangle the performance gains of the proposed method from the implicit knowledge distilled from the powerful judge model, which the ablation study (Table 4) shows is responsible for a significant performance boost.

**Questions:**

1. Can the authors provide an analysis or citation to substantiate the key motivational claims? Specifically, can you show examples where these models fail because they "uncritically accept initial visual interpretations" but succeed in "correcting textual context"?

2. Can the authors provide an analysis that demonstrate visual rethinking and not just textual rethinking? How does this mechanism differ from the "Forced Rethinking" in VL-Rethinker [1]?

3. What is the performance of the full GThinker framework if a smaller, open-source judge model (e.g., Llama 3 70B or an equivalent) is used for the Judge-Guided Selective Training, instead of GPT-4o?

4. In the data pipeline ablation (Table 3), what model was used to generate the data for the 'Image' row, without iterative refinement? Is this data from a single pass of GPT-4o, or a different model?

---
References
1. VL-Rethinker: Incentivizing Self-Reflection of Vision-Language Models with Reinforcement Learning, Wang et al., arXiv 2025

---

### Official Review · Reviewer_roUs · 2025-11-02

**Soundness:** 1
**Presentation:** 3
**Contribution:** 3
**Rating:** 4
**Confidence:** 5

**Summary:**

This paper presents GThinker, an MLLM aimed at addressing the imbalance between strong textual reasoning abilities - such as self-reflection and revision - and the weak or absent re-evaluation of visual interpretations in current models, even when these lead to logical inconsistencies. The core idea is Cue-Rethinking - an adaptive reasoning mechanism that grounds inference in visual cues while strategically revisiting and revising them to resolve inconsistencies and strengthen overall reasoning in MLLMs.

The authors train GThinker through a two-stage process, stage 1: pattern-guided cold start followed by judge-guided selective training and stage 2: incentive reinforcement learning. They introduce GThinker-11k, with 7K cue-annotated chain-of-thought data and 4K diverse reinforcement samples curated using the designed iterative multimodal annotation pipeline.

Experiments show 81.5% accuracy on M3CoT, outperforming O4-mini and improving by 2.1% on general multimodal reasoning benchmarks, while maintaining competitive mathematical reasoning.

**Strengths:**

1. Clear problem identification and framing: The paper makes a strong conceptual contribution by identifying a key limitation in current MLLMs - their tendency to persist with initial visual interpretations even when subsequent reasoning or new contextual information exposes inconsistencies. Framing this as a visual rethinking problem, and proposing to address it through an adaptive cue-rethinking mechanism, is both novel and well-motivated.

2.  Another notable strength is the pattern-guided cold start design, which uses an iterative multimodal annotation pipeline to synthesize contrastive examples with reasoning annotations (iteratively improved) and incorporating cue-rethinking. The judge-guided selective training enhances data efficiency by learning from both cases when an inconsistency in visual cue leads to a wrong answer or

3. Result: GThinker achieves 81.5% on M3CoT, outperforming open-source models and approaching closed-source performance,

**Weaknesses:**

Overall concern:
While the proposed approach is intuitively sound, many of the claims appear overstated, and the empirical gains are either marginal or dataset-specific. The improvements on M3CoT may largely stem from in-distribution bias (the training dataset details in appendix section A.1), making it difficult to conclude that the method has broad impact or generalizable benefits. If the approach is primarily designed for such settings, this should be explicitly stated and motivated.

1. Unsubstantiated or inconsistent claims/motivations
The paper claims (pg. 3–4, Sec. 3.1) that prior CoT and RLVR models fail to generalize in “settings requiring grounded, visually informed deductions.” However, Table 1 shows that GThinker improves least on commonsense reasoning (arguably the least visually grounded category) compared to science and math tasks, contradicting this motivation.
Additionally, there are inconsistencies in Table 1, specifically, in the “Soc.” (commonsense reasoning) column, InternVL2.5-MPO-8B achieves the highest score (82.6), yet GThinker is bolded despite slightly worse performance.

2. Limited and uneven benchmark improvements
From Table 2, the model shows meaningful gains mainly on MMStar, while improvements on other benchmarks are minor or absent. The comparison in Sec. 4.2 focuses on the weakest fine-tuned baseline (LLaVA-CoT-11B), making the reported gains less convincing when stronger finetuned baselines are considered. The approach, although better than the base model Qwen2.5VL-7B, appears to be on par with existing finetuned models rather than clearly superior, making the superiority claims less convincing.

3. Dataset and generalization concerns
Appendix details suggest that most training data are derived from M3CoT, with limited inclusion of other datasets (e.g., ScienceQA). This raises concerns that the performance boost on M3CoT is due to in-distribution overlap rather than genuine generalization. The lack of improvement on out-of-distribution benchmarks further reinforces this bias.

4. Missing evaluation on hallucination robustness
Since the core claim is about improving the model’s ability to “rethink visual cues,” it would be valuable to evaluate on hallucination-focused benchmarks such as HallusionBench [1] or similar datasets. This would better demonstrate whether the proposed method truly enhances visual grounding and reduces hallucinated reasoning.

5. Is this technique model specific or any other base model if trained in the above manner can show similar traits, for instance InternVL-2.5-8B if finetuned using the proposed dataset & approach, does it perform better than other finetuned variants of InternVL2.5-8B ?


[1] Guan, Tianrui, et al. "Hallusionbench: an advanced diagnostic suite for entangled language hallucination and visual illusion in large vision-language models." Proceedings of the IEEE/CVF Conference on Computer Vision and Pattern Recognition. 2024.

**Questions:**

Questions are mentioned in the weakness section.

Overall the work requires
1. Stronger empirical validation - improvement on OOD test sets, hallucination test sets, possibly data scaling tests ?
2. Motivation better aligned with the current empirical results


If the authors provide insights to my points raised in the
a) weakness sections
b) conduct additional experiments on hallucination benchmarks
c) conduct experiments on a different base model

with the aim to solidify empirical validity and generalisability of the approach, i am happy to bump my score.

---

### Note · Authors · 2025-11-14

I have read and agree with the venue's withdrawal policy on behalf of myself and my co-authors.